# Population adaptation in efficient balanced networks

Gabrielle J Gutierrez[1,2†*], Sophie Denève[2‡]

[1]Department of Applied Mathematics, University of Washington, Seattle, United States; [2]Group for Neural Theory, École Normale Supérieure, Paris, France

**Abstract** Adaptation is a key component of efficient coding in sensory neurons. However, it remains unclear how neurons can provide a stable representation of external stimuli given their history-dependent responses. Here we show that a stable representation is maintained if efficiency is optimized by a population of neurons rather than by neurons individually. We show that spike-frequency adaptation and E/I balanced recurrent connectivity emerge as solutions to a global cost-accuracy tradeoff. The network will redistribute sensory responses from highly excitable neurons to less excitable neurons as the cost of neural activity increases. This does not change the representation at the population level despite causing dynamic changes in individual neurons. By applying this framework to an orientation coding network, we reconcile neural and behavioral findings. Our approach underscores the common mechanisms behind the diversity of neural adaptation and its role in producing a reliable representation of the stimulus while minimizing metabolic cost.

DOI: https://doi.org/10.7554/eLife.46926.001

*For correspondence:
ellag9@uw.edu

Present address: †Department of Applied Mathematics, University of Washington, Seattle, United States of America; ‡Département D'Études Cognitives, École normale supérieure, Paris, France

Competing interests: The authors declare that no competing interests exist.

## Introduction

The range of firing rates that a sensory neuron can maintain is limited by biophysical constraints and available metabolic resources. Yet, these same neurons represent sensory inputs whose strength varies by orders of magnitude. Seminal work by *Barlow (1961)* and *Laughlin (1981)* demonstrated that sensory neurons in early processing stages adapt their response threshold and gain to the range of inputs that they recently received. A particularly striking example of such gain modulation at the single cell level has been shown in the fly H1 neuron (*Brenner et al., 2000*). Gain adaptation has been observed in other early sensory circuits (*Blakemore and Campbell, 1969*; *Fairhall et al., 2001*; *Solomon and Kohn, 2014*; *Wark et al., 2007*), such as in the retina (*Kastner and Baccus, 2014*), auditory hair cells (*Nagel and Doupe, 2006*; *Wen et al., 2009*) and is also present in later sensory stages (*Adibi et al., 2013*; *Wainwright, 1999*). Moreover, cortical neurons acquire this property during development (*Mease et al., 2013*).

The work of Laughlin and Barlow was instrumental in uncovering a principle of neural adaption as maximizing information transfer. However, the natural follow-up question concerns the decoding of such neural responses after they have been subject to adaptation. Indeed, such changes in neural gains may result in profound changes of the mapping of neural responses to stimuli in a history-dependent manner. This raises the issue of how such adapting responses are interpreted by downstream sensory areas (*Seriès et al., 2009*; *Webster, 2011*).

One possibility, of course, is that downstream areas do not change their decoding strategy, thus introducing systematic biases in perception. This has been interpreted as the source of perceptual illusions such as the tilt after-effect or the waterfall illusion (*Barlow and Hill, 1963*; *Wainwright, 1999*; *Clifford, 2014*; *He and MacLeod, 2001*; *Schwartz et al., 2009*). Such illusions are classically triggered by long presentations of particularly strong or repetitive stimuli (*Maffei et al.,*

**eLife digest** Humans see, hear, feel, taste and smell the world as spiking electrical signals in the brain encoded by sensory neurons. Sensory neurons learn from experience to adjust their activity when exposed repeatedly to the same stimuli. A loud noise or that strange taste in your mouth might be alarming at first but soon sensory neurons dial down their response as the sensations become familiar, saving energy.

This neural adaptation has been observed experimentally in individual cells, but it raises questions about how the brain deciphers signals from sensory neurons. How do downstream neurons learn whether the reduced activity from sensory neurons is a result of getting used to a feeling, or a signal encoding a weaker stimulus? The energy saved through neural adaptation cannot come at the expense of sensing the world less accurately. Neural networks in our brain have evidently evolved to code information in a way that is both efficient and accurate, and computational neuroscientists want to know how. There has been great interest in reproducing neural networks for machine learning, but computer models have not yet captured the mechanisms of neural coding with the same eloquence as the brain.

Gutierrez and Denève used computational models to test how networks of sensory neurons encode a sensible signal whilst adapting to new or repeated stimuli. The experiments showed that optimal neural networks are highly cooperative and share the load when encoding information. Individual neurons are more sensitive to certain stimuli but the information is encoded across the network so that if one neuron becomes fatigued, others receptive to the same stimuli can respond. In this way, the network is both responsive and reliable, producing a steady output which can be readily interpreted by downstream neurons.

Exploring how stimuli are encoded in the brain, Gutierrez and Denève have shown that the activity of one neuron does not represent the whole picture of neural adaptation. The brain has evolved to adapt to continuous stimuli for efficiency at both the level of individual neurons and across balanced networks of interconnected neurons. It takes many neurons to accurately represent the world, but only as a network can the brain sustain a steady picture.

DOI: https://doi.org/10.7554/eLife.46926.002

*1973*). However, adaptation deeply affects neural responses even at short time scales or after only one repetition of the same stimulus (*Patterson et al., 2013*).

Adaptation could make it impossible to recognize a visual object independently of the stimuli presented previously. An example is given in *Figure 1* where we present successive visual patterns to a population of randomly connected leaky integrate-and-fire (LIF) neurons. For simplicity and for the sake of illustration, the network takes a 7-dimensional time-varying input interpreted as a spatio-temporal sequence of digital numbers (*Figure 1b*, top row). An optimal linear decoder was trained to reconstruct the patterns from the spike counts during the presentation of the patterns. Not surprisingly, the decoder could reconstruct the patterns accurately, regardless of their place in the sequence (*Figure 1b*, 2nd row). We then tested the network in the presence of spike-based adaptation in the LIF neurons. Spike-based adaptation was induced by temporarily hyperpolarizing the neurons after each spike. The time scale of this adaptation was chosen to be long enough to cover several visual patterns. When subjected to this spike-time dependent adaptation, the responses became strongly history dependent, resulting in a highly inaccurate decoding (*Figure 1b*, 3rd row). This would suggest that activity in downstream areas and perceptual interpretations should be based not only on the current sensory responses, but also on the recent history of neural activity (*Fairhall et al., 2001*; *Borst et al., 2005*). In this study, we show that this is not necessarily the case. Recurrent connections can be tuned such that spike-dependent adaptation will not impair the stability of the representation (*Figure 1b*, bottom row).

## Results

### Neural network solving a global cost-accuracy tradeoff

We will start from an objective function quantifying the efficiency of a population of spiking neurons in representing a time varying sensory stimulus, $\phi(t)$. We will then show that appropriate recurrent connections between the neurons, namely connections that maintain a tight balance between the excitation and inhibition received by each neuron, will minimize this objective function and thus, maximize the efficiency of the neural code. For the sake of illustration, we hereby assume that the stimulus is unidimensional and positive, as for luminance or color saturation (see Materials and methods for multidimensional stimuli), and the stimulus has arbitrary units. The stimulus will be decoded from the firing activity of the neurons by summing their responses with their respective readout weights , $w_i$.

$$\hat{\phi}(t) = \sum_i w_i r_i(t) \tag{1}$$

The neural response, $r_i(t)$, is defined as the spike train integrated at a short time scale,

$$\dot{r}_i = -\frac{1}{\tau} r_i + o_i \tag{2}$$

where $o_i(t)$ corresponds to the spike train of neuron $i$. The readout weight of neuron $i$ is denoted as $w_i$ and it is a fixed parameter. One may choose to include a wide range of readout weights in the network. The output estimate, $\hat{\phi}(t)$, can be interpreted as a postsynaptic integration of the output spike trains of the population, weighted by synaptic weights **w**.

We wish to construct a network that will minimize the difference between $\phi(t)$ and $\hat{\phi}(t)$, ensuring an accurate representation of the stimulus. Additionally, we wish to impose, not only accuracy, but also cost efficiency in the neural representation. For biological neurons, spiking comes with inherent metabolic costs. For example, resources are expended after each spike and neurons or neural populations may need some time to recover from a period of strong activity. Albeit many different types of cost can be incorporated into our approach, here we summarize these constraints as a cost term representing the sum of all squared firing rates. Thus, we define an objective function composed of two terms, one representing the precision of the representation, and the other the cost of neural activity (**Boerlin et al., 2013**):

$$E(t) = (\phi(t) - \hat{\phi}(t))^2 + \mu \sum_i f_i(t)^2 \tag{3}$$

Where $f_i(t)$ is the firing history of neuron $i$ and the parameter $\mu$ weights the relative contributions of error and metabolic costs. The firing history, $f_i(t)$, is defined as the spike train integrated with a time constant, $\tau_a$.

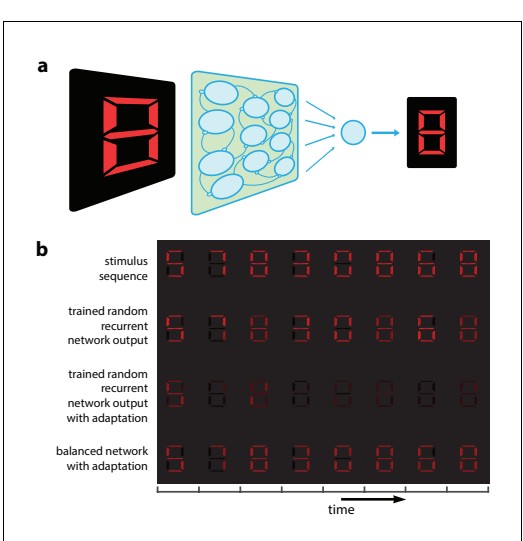

**Figure 1.** Digital number encoding network. (**a**) Schematic of a 7-dimensional input (one dimension for each bar position of a digital interface) being presented to a random recurrent network that sends input to a readout layer (here represented by a single neuron). (**b**) Top, a sequence of digits that serve as stimuli (presented for 200 ms each, spaced by 100 ms between digits). Second row, decoded output of random recurrent network with optimal decoder (trained on 100 samples of completely random patterns). Third row, decoded output of same random recurrent network as above but with adapting neuron responses. Bottom row, balanced network with adaptation derived from efficient coding framework. [All rows: 400 neurons, $\tau = 5ms$ for neural responses integrated by decoder; 3rd and 4th row: $\mu = 0.02; \tau_a = 2000ms$ for the adaptive firing rates].
DOI: https://doi.org/10.7554/eLife.46926.003

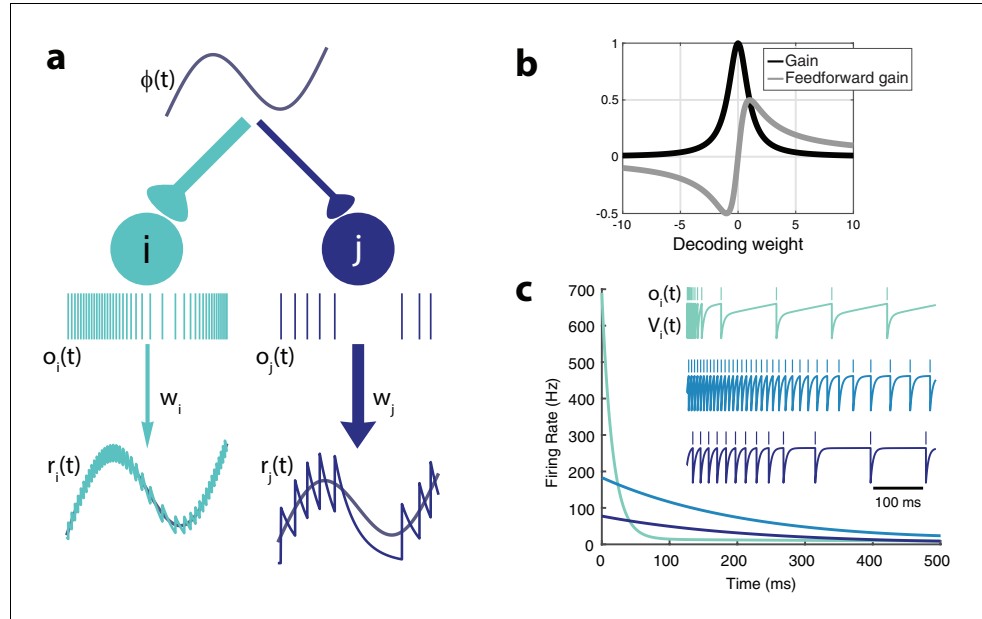

**Figure 2.** Intrinsic model neuron properties. (**a**) High gain neurons (light blue) are intrinsically excitable and due to their small decoding weights they are precise while low gain neurons (dark blue) are less excitable and less precise. An arbitrary input, $\phi(t)$, elicits distinct responses from the two neurons (spikes train $o_i$ and $o_j$, respectively). Neurons send a filtered response, $r_i$, $r_j$, to the decoder weighted by $w_i$ and $w_j$, respectively. (**b**) Relationship between gain $g_i = 1/(w_i^2 + \mu)$, feedforward gain $g_i w_i = w_i/(w_i^2 + \mu)$, and decoding weight $w_i$ ($\mu = 1$). (**c**) Different gains give neurons distinct adaptation dynamics. Instantaneous spiking rates in response to a constant input are plotted over time for three model neurons with different decoding weights (light blue, w = 1; medium blue, w = 5; dark blue, w = 9). High gain neurons have the steepest adaptation (light blue) whereas low gain neurons (dark blue) do not adapt as rapidly given the same input. Inset shows the voltage trace, $V(t)$, and spike train, $o(t)$, for each example neuron.

DOI: https://doi.org/10.7554/eLife.46926.004

$$\dot{f}_i = -\frac{1}{\tau_a} f_i + o_i \qquad (4)$$

Typically, the adaptation time scale is assumed to be significantly longer than the decoder time scale ($\tau_a \gg \tau$). A short $\tau$ (e.g. of the order of 10 ms) ensures that fast changes in the stimulus can be represented accurately. However, the metabolic cost of spiking accumulates and recovers at slower time scales (e.g. $\tau_a$ corresponds to hundreds of ms). The underlying assumption is that the dynamics allowing metabolic resources to be replenished are slower than the time scale at which neural populations transmit information. The sum of squared firing history will encourage, not only low activity at the level of the population, but also low activity in single neurons. As a result, neurons will share the burden of the representation.

From these assumptions, we derive a prescription for the voltage dynamics of leaky integrate-and-fire (LIF) neurons performing a greedy minimization of the objective function, $E$ (see Materials and methods for full derivation). Our framework revolves around the assumption that a neuron spikes only when doing so reduces the decoding error. This condition can be expressed in terms of the objective function as $E(t)^{no \,>spike} > E(t)^{spike}$ where a spike is justified if the objective is minimized relative to having no spike at that time step. Using *Equation 3*, we obtain a new expression from this inequality that embodies a condition for spiking and that we interpret as a voltage expression and a threshold (see Materials and methods for derivation details) such that $V(t) > threshold$ and voltage is:

$$V_i(t) = \frac{1}{w_i^2 + \mu}(w_i(\phi(t) - \hat{\phi}(t)) - \mu f_i(t)) \tag{5}$$

Taking the derivative of the voltage expression produces the voltage equation below:

$$\tau \dot{V}_i = -V_i + g_i w_i(\tau \dot{\phi} + \phi) - \tau g_i \sum_j \Omega_{ij} o_j - \kappa_i f_i \tag{6}$$

Where $g_i$ is the gain of neuron $i$,

$$g_i = 1/(w_i^2 + \mu) \tag{7}$$

and the lateral connections are given by $\Omega_{ij}$

$$\Omega_{ij} = w_i w_j + \mu \delta_{ij} \tag{8}$$

where $\delta_{ij}$ is the delta function (equals one only if $j = i$, zero otherwise) and $\kappa_i = \mu g_i(1 - \frac{\tau}{\tau_a})$.

The form of the voltage equation is amenable to being interpreted as a set of currents to a neuron embedded in a recurrent network with all-to-all connectivity. Neurons are connected by mutually inhibitory synapses determined by their decoding weights. The final term corresponds to an adaptation current that depresses the voltage as a function of its recent activity (see *Figure 2c*). This indicates that spike-frequency adaptation in single neurons is part of the solution to the cost-accuracy tradeoff. However, we will show that it cannot work alone; it needs to be associated with appropriately tuned recurrent connections.

It is easier to interpret the network function if we consider that the membrane potentials are effectively proportional to the global coding error penalized by the past activity of the neuron, as seen in *Equation 5*. A neuron that reaches the firing threshold is guaranteed to contribute a decrease of the error term in the objective function (*Equation 3*). As a whole, the population performs a greedy minimization of the objective function, or, in other terms, a greedy maximization of the coding efficiency.

Finally, we note that since the integrated excitatory input, $g_i w_i \phi(t)$, is cancelled as precisely as possible (except for the cost penalty) by the recurrent inhibition, $-g_i w_i \hat{\phi}(t)$, the network can be considered as balancing feedforward excitation and recurrent inhibition (see *Equation 5*). The second ingredient for population efficiency (in addition to spike-based adaptation) is thus to maintain a tight E/I balance in the network. In other words, we show that a memoryless decoder will be able to reconstruct the stimulus from the output spike trains of an E/I balanced population of adapting neurons. This is shown in the bottom row of *Figure 1b*. Before we investigate the network dynamics and performance, we first describe the properties of single neurons and the relationship between their gain and their coding precision.

## Single neuron properties

Let us first consider the case without quadratic cost (i.e. $\mu = 0$). In that case, each neuron effectively has identical voltage and spiking dynamics. Neurons are differentiated only by their gain, $g_i$, and their decoding weight, $w_i$ (*Figure 2*). The strength of the feedforward gain is inversely related to the strength of the output weight for each neuron. As a result, neurons with the smallest decoding weights (and thus, the highest precision in representing the input) tend to respond most strongly to the stimulus (*Figure 2a*). We will refer to these costly but reliable neurons as 'strongly excitable'. In contrast, neurons with large decoding weights and small input weights (thus 'low gain' neurons) bring less precision to the estimate but are metabolically efficient. We will refer to these neurons as 'weakly excitable'.

Note that if $\mu = 0$, the cost is not taken into account by the network. Thus, it will always favor precision over cost. In that case, only the most excitable neuron (with the smallest decoding weight) will respond to the stimulus while completely inhibiting the other neurons. However, with the addition of a cost ($\mu > 0$), adaptive currents contribute to the voltage dynamics, penalizing neurons with large firing rates. Moreover, the feedforward gain, $g_i w_i$, does not necessarily decrease monotonically with the decoding weight (*Figure 2b*). For very small decoding weights, $|w_i|^2 << \mu$, neurons with decoding weights smaller in magnitude than $\sqrt{\mu}$ are penalized. These neurons would simply be too costly to

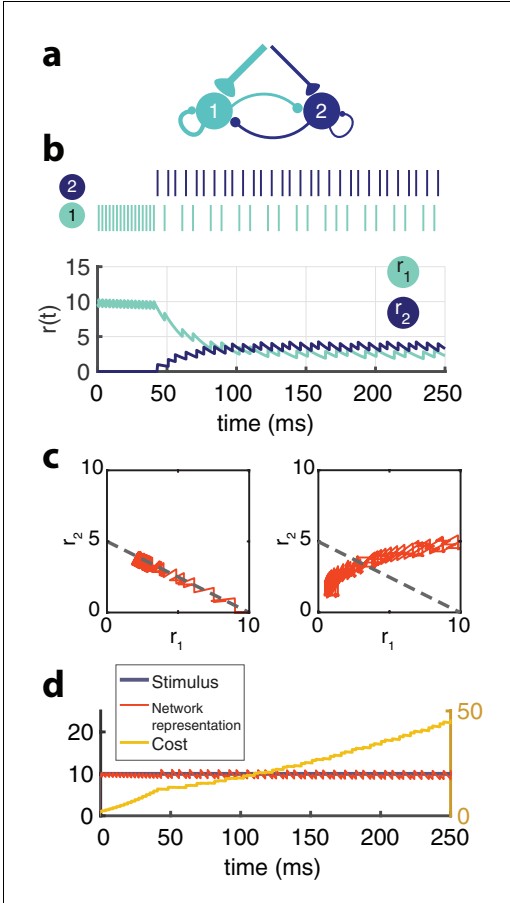

**Figure 3.** Two-neuron network. (a) Schematic of recurrently connected two-neuron network derived from efficient coding framework. Neuron 1 is strongly excitable ($w_i = 1$), while neuron 2 is weakly excitable ($w_i = 2$). (b) Spikes from neuron 1 (light blue) and neuron 2 (dark blue) show the transient response of the strongly excitable neuron and the delayed, but sustained response of the weakly excitable neuron (top) in response to a constant stimulus. Postsynaptic activity, $r(t)$ (bottom) [$\mu = 0.02, \tau = 25ms, \tau_a = 1000ms, \phi(t) = 10$]. (c) The balanced network with adaptation follows a linear manifold (left), whereas the network without recurrent connections but with adaptation cannot be linearly decoded (right). (d) The cost ($\mu \sum_n f_n^2$, yellow) accumulates steeply until neuron one adapts and neuron two is recruited and the cost increases at a slower rate. The network representation (orange) is maintained despite the redistribution of activity among the neurons.

DOI: https://doi.org/10.7554/eLife.46926.005

participate meaningfully in the cost/accuracy tradeoff solved by the population. Model neurons in isolation (i.e. without any contribution from recurrent connections) would respond to a step-like input with a rate that decreases exponentially in time before reaching a plateau (*Figure 2c*), a classic signature of activity-dependent suppression. The time constant of this adaptation is determined by $\tau_a$, while the strength of this adaptation increases with the gain. Highly excitable neurons adapt strongly, while less excitable neurons adapt weakly.

However, these intrinsic properties of single neurons will be deeply affected by the dynamics introduced by recurrent connections. To gain a better understanding of population adaptation, we investigate how inhibitory connections orchestrate the relative contributions of different neurons over the duration of a long stimulus.

## Network activity is distributed on a manifold in neural activity space

We first illustrate the effect of recurrent connections with an example network composed of only two neurons (*Figure 3*). The two neurons are reciprocally connected with inhibitory connections, as prescribed in the derivation (schematized in *Figure 3a*). They receive a constant stimulus, but have different input weights.

During sustained stimulation, the response of each neuron fluctuates dynamically despite the fact that the stimulus is constant (*Figure 3b*). This would be expected given their spike-time dependent adaptation. However, if one removes the recurrent connections and plots the response of one neuron as a function of the other (*Figure 3c*, right), we discover that the population response wanders from the iso-coding line, $\hat{\phi} = w_1 r_1 + w_2 r_2$ (i.e. the manifold in activity space where the stimulus would be decoded properly). In contrast, the intact network with its recurrent connections coordinates the two neurons such that the weighted sum of their responses remains accurate. The movement of the activity along the manifold defined by the constant stimulus and the decoding weights reflects a progressive redistribution of activity to satisfy the unfolding cost-accuracy tradeoff, as the cost slowly accumulates (*Figure 3d*).

While to a naive observer, the high gain neuron may appear to adapt while the low gain neuron has a sustained response and a longer delay, in fact both contribute to population adaptation because both neurons coordinate and adapt their activity to limit the metabolic cost of the representation while maintaining its accuracy. Recurrent

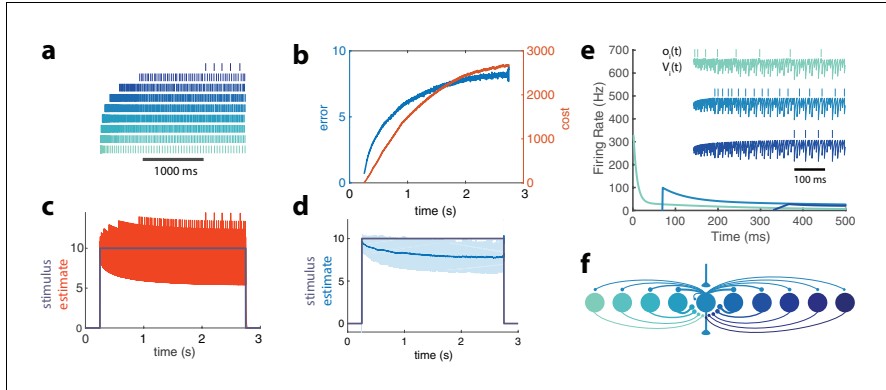

**Figure 4.** Adapting population of heterogeneous neurons. (**a**) Spike raster of all 10 neurons in a balanced network with adaptation in response to a pulse stimulus ($\mu = 0.2, \tau = 5, \tau_a = 1000, \phi(t) = 10, w = [1, 2, ..., 10]$). Neurons are ordered from weakly excitable (top, dark blue) to highly excitable (bottom, light blue). (**b**) Both the error ($(\phi(t) - \hat{\phi}(t))^2$, blue) and cost ($\mu \sum_n f_n^2$, orange) accumulate over time. (**c**) The network estimate ($\hat{\phi}(t)$, orange) tracks the stimulus ($\phi(t)$, gray) with increasing variance. (**d**) The smoothed network estimate (blue line) shows a biased estimate with increasing variance (blue shade, standard deviation). (**e**) Instantaneous spiking rates of 3 example neurons in the network. Inset shows the voltage trace, $V(t)$, and spike train, $o(t)$, for each example neuron. (**f**) Schematic of 10-neuron balanced network showing only connections to and from the middle neuron. Excitatory connections are shown as triangles and in this particular network are only found in the feedforward and output connections. Inhibitory connections are shown with small circles and make up only the recurrent connections.

DOI: https://doi.org/10.7554/eLife.46926.006

connections deeply affect the dynamics of each neuron. For example, the inhibition from the strongly excitable neuron is responsible for the response delay of the weakly excitable neuron.

## Coordinated adaptation in a neural population

Within a network with many neurons (*Figure 4*), recurrent connections interact with the intrinsic properties of the neurons in a similar manner as in the previous example. The first neurons to be recruited are strongly excitable and provide an initially very precise representation of the signal.

These neurons inhibit the less excitable neurons, preventing them from firing early in this stimulation period. As the cost accumulates, however, the response of the high gain neurons decays due to spike-frequency adaptation. This is compensated by weakly excitable neurons that become disinhibited, fire, and then adapt in their turn. The less excitable a neuron is, the later it will be recruited, resulting in strong response delays. The dynamic response properties of individual neurons are thus dominated by network interactions and are markedly different from their intrinsic adaptive properties (*Figure 2*).

Because the disinhibition of weakly excitable neurons automatically compensates for the decay in strongly excitable neural responses, the stimulus representation remains stable during the whole period (*Figure 4d*). However, note that its precision degrades as more low gain neurons contribute to the representation. As a result, the bias and standard deviation of the representation increases as imposed by the global cost/accuracy tradeoff.

## Coordinated adaptation of tuning curves

To illustrate what coordinated adaptation would predict for tuning curves measured experimentally, we constructed a population of neurons that code for visual orientation; V1 simple cells. The input to the network takes the form of a two-dimensional signal with a cosine and a sine of the presented orientation (see Materials and methods). Each neuron has a preferred orientation that is given by the combination of input weight strengths in the two input dimensions. In turn, the network orientation estimate can be decoded from the population (see Materials and methods).

The lateral connections derived from the model maximally inhibit neurons with similar preferred orientations and excite neurons with orthogonal orientations (see schematic in *Figure 5a*) due to the

choice of decoding weights which can be positive or negative, or some combination. To observe the effects of adaptation on a diverse population of neurons, we constructed our network so that neurons have equally spaced preferred orientations and a partner neuron that shares the same preference but has a different gain. There is a high gain and a low gain neuron among each pair of neurons that code for the same orientation.

*Figure 5b* illustrates the spiking response of the network to a prolonged oriented stimulus. As seen in the simpler model from *Figure 4*, high gain neurons respond first, then adapt. As the responses of those strongly excitable neurons decay, weakly excitable neurons are recruited, maintaining the representation. This results in systematic changes in the tuning curves from early in the response to later in the response (*Figure 6*). Highly excitable neurons are suppressed relative to their early responses (*Figure 6*, top). In contrast, weakly excitable neurons see their tuning curves widen when the adapting stimulus is similar to their preferred orientation (*Figure 6*, bottom). At the flank of the adapting orientation, low gain neurons see an increase in their responsiveness. Here, the network interactions override the intrinsic adapting currents in the weakly excitable neural population. In other words, the disinhibition from strongly excitable neurons combined with the constant feedforward drive to these low gain neurons results in facilitated activity rather than the suppressed activity one would expect to be caused by adaptation. Finally, the tuning curves for the most excitable neurons are broader than those for weakly excitable neurons. These neurons are more likely to fire first in response to oriented stimuli that are near their preferred orientation and prevent the low gain neurons from doing the same.

## Heterogeneity in a more plausible model

The same qualitative effects are observed in a more realistic network where the preferred orientations of different neurons and their decoding weights are taken from a random distribution (*Figure 7*), rather than regularly spaced with two levels of excitability. The tuning curves are more heterogeneous not because of noise but because of the randomness of the decoding weights. Tuning curves can be either facilitated or suppressed by adaptation. When the adapted stimulus falls on the flank of the tuning curve, it can be accompanied by a shift toward or away from the adapting stimulus. The effect of adaptation on single neurons is variable not because of noise (we did not introduce any) but because of local heterogeneity in the competition they receive from other neurons, itself due to the random choices of weights. In fact, adaptation in one neuron would be impossible to predict quantitatively without observing the rest of the network.

## Perceptual adapation

We have stressed the accuracy of the stimulus representation in the face of time-varying activity due to adaptation. While this kind of activity could be interpreted as leading to a stable percept in spite of adaptation, we acknowledge that perceptual errors and biases are abundant in the natural world. Our network is capable of emulating these errors and it is able to do so in a manner that is consistent with experimental findings. The network is designed to negotiate the tradeoff between accuracy and efficiency and it will prioritize the production of a stable representation if $\mu$ is small. If $\mu$ is large, the network will favor cost over accuracy. Thus, strong adaptation and a prolonged stimulus presentation can produce a representation that degrades over time. This degradation can lead to a bias in the decoder. In *Figure 8*, an oriented, strong, adapting stimulus is presented for 2 seconds followed by a test orientation (this is schematized in *Figure 8a*). An example of the resulting network activity is shown in *Figure 8b*.

Before adaptation takes hold, the adapting stimulus activates the high gain neurons that have preferences at or near the stimulus orientation. Because the adapting stimulus is strong, high gain neurons with similar preferences are quickly recruited. As the stimulus persists, the most strongly activated high gain neurons fatigue and the low gain neurons with matching preferences are recruited. After the presentation of the adapting orientation, a weaker peripherally oriented test stimulus is delivered. The response distribution and dynamics are markedly different. Instead of a widely-tuned response, the weaker test stimulus produces a more narrowly distributed response. The decoded orientation is offset from the test stimulus orientation, indicating a bias in the perceived orientation.

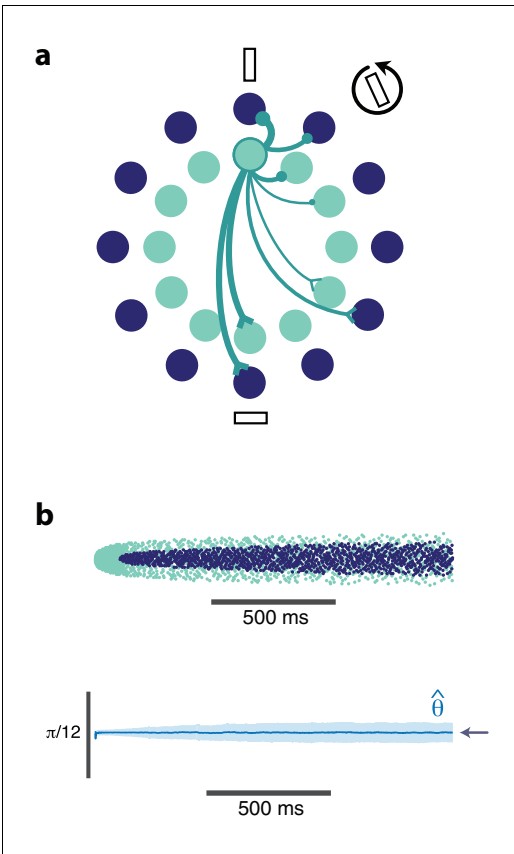

**Figure 5.** Orientation coding-network. (**a**) Schematic showing the dual-ring structure of the network of high gain (light blue) and low gain (dark blue) neurons. Some of the recurrent connections from the outlined light blue neuron are illustrated to show that a neuron inhibits its neighbors most strongly and excites neurons with opposing preferences (inhibitory connections are shown as circles, excitatory connections are shown as chevrons). (**b**) Spike raster (top) of population activity showing the evolution of the population response during a prolonged stimulus presentation of a constant orientation. Rasters are displayed in order of neuron orientation preferences. The decoded orientation is steady while the variance increases over time (bottom). Arrow indicates the stimulus orientation. ($\mu = 0.1, \tau = 5ms, \tau_a = 2000ms, \eta = 10$, stimulus magnitude C = 50, 200 neurons).

DOI: https://doi.org/10.7554/eLife.46926.007

A classical study of perceptual bias is the tilt illusion (*Gibson and Radner, 1937*; *Clifford, 2014*). In the tilt illusion, the orientation of a test grating is perceived incorrectly after adaptation to a differently oriented stimulus. Experimental studies report that the perceived orientation is often repulsed away from the adapted orientation, the effect being maximal for adapting stimuli tilted around 15–20 degrees from the test stimulus. If the adapting stimulus is oriented around 60 degrees from the test stimulus, a repulsive effect is observed instead. This effect has been confirmed in the visual cortex (*Jin et al., 2005*; *He and MacLeod, 2001*). Our model replicates this effect (*Figure 8c,d*). The test stimulus is decoded at an orientation that is repulsed from its actual orientation away from the adaptor when the adaptor is approximately 15 degrees from vertical (*Figure 8c*, middle). However, when the adaptor is obliquely oriented from the test orientation, the test stimulus is perceived to be oriented in a direction that is attracted to the adaptor (*Figure 8c*, right). Test stimuli within a range of 0–45 degrees difference from the adaptor orientation are repulsed whereas test stimuli with a greater than 45 degree difference from the adaptor orientation are attracted (*Figure 8d*). In accordance with experimental findings, the repulsion effect has a greater amplitude than the attraction effect.

## Discussion

Sensory neurons in cortex are embedded in highly recurrent networks with each cell receiving strong inhibitory currents that co-vary with excitatory currents (*Graupner and Reyes, 2013*) and are thus E/I balanced. Here, we show that in balanced networks, heterogeneous sensory neurons with activity dependent suppression solve a global cost/accuracy tradeoff rather than a local tradeoff at the level of each neuron. In our case, adaptation at the level of individual neurons co-exists with a largely stable representation at the population level. Rather than being globally suppressed by adaptation, E/I balance indirectly ensures that the neural activity is redistributed from highly responsive neurons to less responsive neurons without changing the interpretation of this activity by downstream areas.

Our approach suggests that, given adaptation, neural coding cannot be understood at the level of a single neuron, except in cases where a unique sensory feature is solely encoded by a single neuron, such as the H1 neuron (*Brenner et al., 2000*). In areas containing large numbers of interconnected neurons with redundant selectivity, many questions about neural coding and adaptation are only meaningful when applied to whole populations. We show that the adapting tuning curves of a single neuron can reflect a collective, flexible solution found by the network in particular contexts. Other studies have used predictive coding and efficient coding frameworks to construct models of

adapting neural responses (*Chopin and Mamassian, 2012*; *Mlynarski and Hermundstad, 2018*; *May and Zhaoping, 2016*) , however, our model incorporates a biologically plausible spike-frequency adaptation mechanism.

Another problem arises in the trial-to-trial variability produced by adaptation that is observed at the single neuron level. As mentioned in the introduction, the history-dependence caused by adaptation begs the question of how a consistent representation can be decoded from a network in which all, or most, neurons are subject to adaptation. Our study shows that the potentially harmful effects of adaptation on the individual neuron's ability to encode a stimulus can be mitigated by a coordinated population response. Other studies that have addressed this issue propose updating the decoder (*Benucci et al., 2009*) or have considered divisive gain control mechanisms (*Schwartz et al., 2009*) as well as synaptic plasticity mechanisms (*Hosoya et al., 2005*). Our study offers an alternative, plausible framework for resolving the cost-accuracy tradeoff on a shorter time scale than the operating time scale for synaptic plasticity. Instead of updating the weights to better represent stimuli over several iterations, as is done for the perceptron and convolutional neural networks (*Fukushima, 1980*; *LeCun et al., 1999*; *Olshausen and Field, 1996*; *Rosenblatt, 1958*), we derive a prescription for the voltage dynamics so that the network neurons can produce a reconstruction of any stimulus with fixed decoder and recurrent weights.

Our model is developed from a normative encoding framework (*Boerlin et al., 2013*; *Druckmann and Chklovskii, 2012*; *Olshausen and Field, 1996*; *Spratling, 2010*) in which we enforce efficiency in the encoder and accuracy in the decoder. The new contribution compared to *Boerlin et al. (2013)* is to allow more flexibility in the form that the metabolic cost can take. In particular, the time scales of the cost and of the representation are disassociated which leads to distinct dynamics for the cost and for the neural firing rate. This approach can be generalized to many other types of cost, arbitrary weights, and number of neurons.

## Single neuron coding is dynamic rather than a static property

Our model suggests that diverse adaptation properties within a population can be an asset. The variability of adaptation effects has been observed in V1 neurons (*Jeyabalaratnam et al., 2013*; *Nemri et al., 2009*; *Ghisovan et al., 2009*). A heterogeneous population of neurons is able to better distribute the cost to maximize efficiency in different contexts. Studies in the retina show that retinal ganglion cells with different adaptive properties complement each other such that sensitizing cells can improve the encoding of weak signals when fatiguing cells adapt (*Kastner and Baccus, 2011*). This arrangement is particularly advantageous for encoding contrast decrements which would be difficult to distinguish from the prior stimulus distribution if only suppressive adaptation prevailed. At the same time, these heterogeneities contribute to complex dynamics in the neural spike trains (*Dragoi et al., 2000*; *Okun et al., 2015*; *Nirenberg et al., 2010*; *Mohar et al., 2013*; *Wissig and Kohn, 2012*), obscuring the relationship between neural activity and neural coding for an observer of single neuron activity. We make the prediction that neurophysiological studies where single neuron activity is recorded may exhibit an experimental bias that results in highly responsive neurons being overrepresented in the sample.

Moreover, our study challenges the notion that tuning is a static characteristic of neurons. Experiments increasingly reveal that neurons change their tuning dynamically with changing stimulus statistics (*Hollmann et al., 2015*; *Hong et al., 2008*; *Hosoya et al., 2005*; *Nagel and Doupe, 2006*; *Smirnakis et al., 1997*; *Solomon and Kohn, 2014*; *Wark et al., 2007*; *Wark et al., 2009*). In the visual cortex, it has been shown that the tilt after effect is not only an effect of response suppression but that it also has the effect of shifting the tuning curves of neurons away from their preferred orientations (*Jin et al., 2005*; *Ghisovan et al., 2009*; *Dragoi et al., 2000*). While it may be possible to predict some aspect of the tuning change from measurements of intrinsic neuron properties, our study shows that a great deal of the change may be a network effect rather than an intrinsic neuronal effect. Thus, the extent of adaptation for a single neuron may be difficult to predict without considering the properties of the rest of the network (*Fairhall, 2014*). Such unpredictable adaptation could be a problem for the interpretation by downstream readouts, however, we show that when the network is considered as a whole, the adaptive effects in one neuron can be compensated for by another neuron that reports to the same readout. In other words, the apparently complex adaptation at the single neuron level is not an impediment to the network but rather an indicator of the manner in which the signal is encoded by the network as a whole.

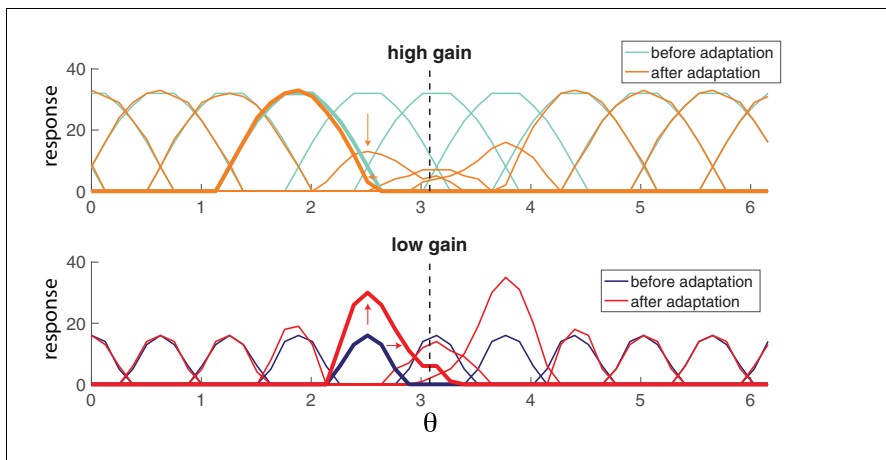

**Figure 6.** Population adaptation tuning curves show neuron responses to a full range of test orientations (x-axis) after adaptation to a single orientation (black dashed line). Top, tuning curves for strongly excitable neurons before adaptation (light blue) are broad. After adaptation (orange), tuning curves near the adaptor are suppressed. Bottom, tuning curves for weakly excitable neurons before adaptation (dark blue) show less activation than for high gain neurons and more specific tuning. After adaptation (red), flanking curves are facilitated and shifted toward adaptor. [$\mu = 0.1, \tau = 5ms, \tau_a = 2000ms, \eta = 10$, stimulus magnitude C = 50, 200 neurons].
DOI: https://doi.org/10.7554/eLife.46926.008

## Validating the framework experimentally

Our model applies at the level of relatively densely connected, and thus local, populations. Observing the organized transfer of responses between neurons through adaptation and E/I balance would require one to record a significant proportion of these neurons locally (neurons that are likely to be interconnected directly or through interneurons). Recent experimental techniques render such recordings possible (*Buzsáki, 2004*), bringing an experimental validation of this framework within grasp. These recordings could be compared before and after adaptation, over the duration of prolonged stimuli, or over many repetitions of the same stimulus. What we expect to see is a generalization of the effect illustrated in *Figure 4b,c* to larger neural populations. First of all, there should exist a decoder of neural activity, independent of stimulus history that can detect the stimulus despite large changes in neural activity over time. Second of all, shuffling the neural responses, for example between the early and latter part of the responses to a prolonged stimulus, should have detrimental effects on such stable decoding. And finally, over the course of adaptation, the activity of the different neurons should not vary independently. For example, if we performed a dimensionality reduction (such as a principle component analysis) of the neural population activity during a prolonged stimulus presentation, we might be able to observe that neural responses over time are constrained on a subspace where the stimulus representation is stable. Another, more direct way of testing our framework would be to activate or inactivate a part of the neural population. This could be done optogenetically, for example (*Okun et al., 2015*).

## Materials and methods

All simulations were done in Matlab using code that we developed from the spiking predictive coding model that is mathematically derived below.

### Digital number encoding network

The network used in *Figure 1* is a generic recurrent network of 400 neurons with random recurrent and feedforward weights. The feedforward weights are a 7 × 400 matrix of values drawn from a uniform distribution in the [−1,1] range. The recurrent weights are drawn from a Gaussian distribution with mean = 0, std = 0.87 (close to 1) and are a 400 × 400 matrix, however, all neurons had an autapse that was the sum of the negative squares of its feedforward weights. The network was trained on 100 stimulus examples of 300 ms each that were generated randomly from a uniform

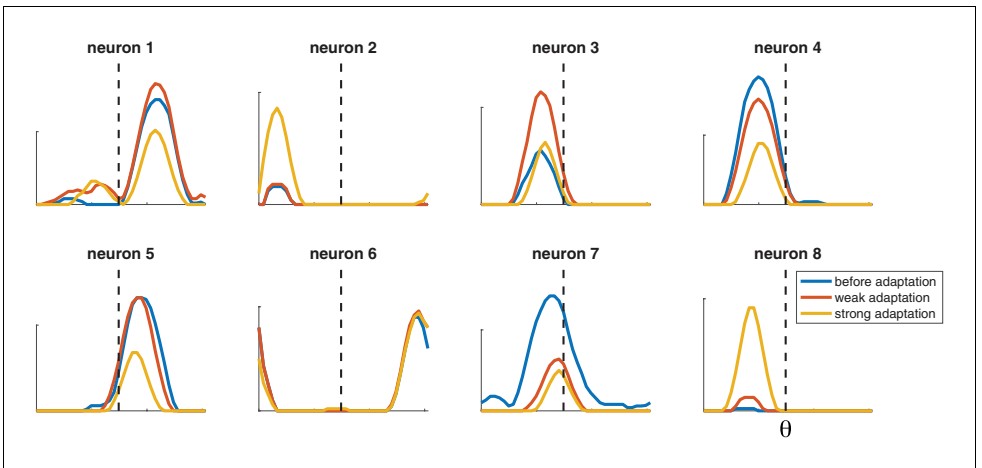

**Figure 7.** Selected tuning curves from orientation network with random decoder weights (and thus random neuron gains). Blue curves, before adaptation; red curves, after weak adaptation; yellow curves, after strong adaptation. Some neuron responses are suppressed after adaptation while others are facilitated, and some tuning curves shift laterally after adaptation. Dashed lines indicate adaptor orientation.
[$\mu = 0.2, \tau = 5ms, \tau_a = 1000ms, \eta = 1$, weak stimulus magnitude C = 10, strong stimulus magnitude C = 50, test stimulus magnitude C = 10, 200 neurons].
DOI: https://doi.org/10.7554/eLife.46926.009

distribution of arbitrary input values between 0 and 4. An optimal linear decoder was obtained from this training by taking the inverse of the responses and multiplying them by the stimulus training examples: decoder = pseudoinverse $(r(t))$. The trained network was then presented with a sequence of 8 digitized patterns for 200 ms each separated by 100 ms of no stimulus input. To demonstrate the effect of adaptation, the trained network was run on the same stimulus sequence and with the same linear decoder but this time the spiking threshold was dynamically regulated by past spiking activity such that the threshold was $1 + \mu f_i(t)$, where $\dot{f}_i(t) = -\frac{1}{\tau_a}f_i(t) + o_i(t)$. For the example of the balanced network with adaptation, the network was derived using the framework described below using the following parameters: $\tau_a = 2000ms, \mu = 0.02, \tau = 5ms$.

## Network model

We provide here a brief description of the network structure and the objective function it minimizes. A detailed and closely related version of this derivation is found in *Boerlin et al. (2013)*. The innovation in our present study is the incorporation of a variable for spiking history in the derivation. We consider a spiking neural network composed of N neurons that encodes a set of M sensory signals, $\phi = [\phi_1, ..., \phi_M]$. Estimates of these input signals, $\hat{\phi} = [\hat{\phi}_1, ..., \hat{\phi}_M]$, are decoded by applying a set of decoding weights, $[w_{i1}, w_{i2}, ..., w_{im}]$, to the filtered spike train of neuron $i$ so that $\hat{\phi}_m(t) = \sum_i^N w_{im} r_i(t)$ (see *Equation 1*). The filtered spike train, $r_i(t)$, corresponds to a leaky integration of its spikes, $o_i(t)$, while the spike history, $f_i(t)$, filters the spike train on a longer time scale so that $\tau_a > \tau$.

$$o_i(t) = \sum_k \delta(t - t_i^k)$$

(9)

$$\dot{r}_i = -\frac{1}{\tau}r_i + o_i$$

(10)

$$\dot{f}_i = -\frac{1}{\tau_a}f_i + o_i$$

(11)

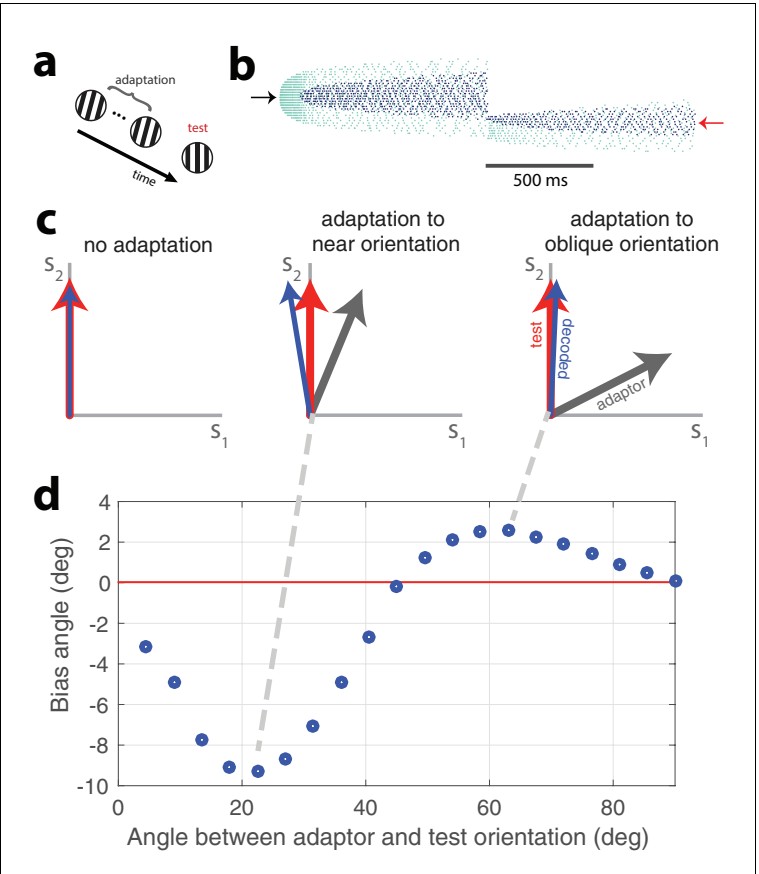

**Figure 8.** Tilt illusion. (a) Schematic of tilt adaptation protocol. (b) Network activity in response to an adapting stimulus followed by a test stimulus. Rasters are ordered by neurons' orientation preferences. Black arrow, neurons that prefer adapting orientation; red arrow, neurons that prefer test orientation ($\mu = 0.1, \tau = 5ms, \tau_a = 2000ms, \eta = 10$, 200 neurons, adaptor C = 50, test C = 25). (c) Examples of tilt bias: (left) no bias before adaptation, (middle) network estimate is biased away from test stimulus and adaptor when adaptor is near test orientation, (right) estimate is biased towards adaptor when adaptor is at large angle to test stimulus (red arrow, test orientation; grey arrow, adaptor; blue arrow, decoded orientation to test orientation after adaptation). (d) Estimate bias is repulsive for near adaptation and attractive for oblique adaptation. Adaptor is presented for 2 s and test orientation is presented for 250 ms ($\eta = 0$, adaptor C = 25, test C = 5).

DOI: https://doi.org/10.7554/eLife.46926.010

with $t_i^k$ the spike time of the $k^{th}$ spike in neuron $i$ and $\tau$ the time scale of the decoder. As we will see, $\tau$ is the membrane time constant of the model neurons and $\tau_a$ is the adaptation time constant.

The decoding weights $w_{im}$ are chosen a priori. They determine the selectivity and gain of the model neurons. We want to construct a neural network that represents the signals most efficiently, given the fixed decoding weights. Efficiency is defined as the minimization of an objective function composed of two terms, one penalizing coding errors, and the other penalizing firing rates:

$$E(t) = \|\phi(t) - \hat{\phi}(t)\|^2 + \mu \sum_{i}^{N} f_i^2 \tag{12}$$

$\mu$ is a positive constant regulating the cost/accuracy tradeoff. In order to minimize this objective function, we define a spiking rule that performs a greedy minimization. Thus, neuron $i$ fires as soon as this results in a minimization of the cost, that is as soon as $E^{spike\ in\ i}(t) < E^{no\ spike\ in\ i}(t)$. A spike in neuron $i$ contributes a decaying exponential kernel, $h(u)$, to its firing rate so that

$$r_i(u) \to r_i(u) + h(u - t) \tag{13}$$

$$\hat{\phi}_m(u) \to \hat{\phi}_m(u) + h(u-t) \tag{14}$$

$$f_i(u) \to f_i(u) + \tilde{h}(u-t) \tag{15}$$

where $\tilde{h}(u)$ is a more slowly decaying exponential kernel than $h(u)$. The spiking condition, $E^{spike\,in\,i}(t) < E^{no\,spike\,in\,i}(t)$, can be expressed as:

$$\|\phi - \hat{\phi} + w_i h e_i\|^2 + \mu \sum_{n \neq i}^{N} f_n^2 + \mu(f_i + \tilde{h})^2 < \|\phi - \hat{\phi}\|^2 + \mu \sum_{n}^{N} f_n^2 \tag{16}$$

The i-th element in the Euclidean basis vector, $e_i$, is one while all other entries are zero. Algebraically rearranging this expression leads to the following spiking rule: neuron $i$ spikes if:

$$g_i\left(\sum_{m}^{M} w_{im}(\phi_m(t) - \hat{\phi}_m(t)) - \mu f_i\right) > \frac{1}{2} \tag{17}$$

$$g_i = 1/(\|w_i\|^2 + \mu) \tag{18}$$

With $g_i$ being the 'gain' of neuron $i$. We interpret the left-hand side of *Equation 17* as the membrane potential, $V_i(t)$, of neuron $i$, and the right-hand side as its firing threshold. Membrane potentials are normalized such that each neuron has a threshold equal to 1/2 and reset potential equal to $-1/2$. The membrane potential dynamics are obtained by taking the derivative of the voltage expression with respect to time (where $\kappa_i = \mu g_i(1 - \frac{\tau}{\tau_a})$):

$$\tau \dot{V}_i = -V_i + g_i \sum_{m}^{M} w_{im}(\tau \dot{\phi}_m + \phi_m) - \kappa f_i - \tau g_i \sum_{m}^{M} \sum_{j}^{N} w_{im} w_{jm} o_j - \mu \tau g_i o_i \tag{19}$$

Notice that the lateral connection between neuron $i$ and neuron $j$ is equal to $\tau g_i \sum_m w_{im} w_{jm}$. Thus, the lateral connections measure to what extent the feed-forward connections of two neurons are correlated, and they remove those correlations to obtain the most efficient code.

## Orientation model

The orientation-coding network follows the same derivation as outlined for the generic network model. It has two input dimensions and 200 neurons. There are two subpopulations of neurons such that 100 neurons are high gain neurons and the remaining 100 neurons are low gain neurons. Both populations span the unit circle evenly such that one low gain and one high gain neuron share the same preferred orientation.

More precisely, we endowed each neuron with a decoding vector , $w_i$:

$$[w_{i1}, w_{i2}] = [\gamma_i \cos(2\Theta_i), \gamma_i \sin(2\Theta_i)], -\frac{\pi}{2} < \Theta_i < \frac{\pi}{2} \tag{20}$$

$\gamma_i$ equals three for high gain neurons and nine for low gain neurons and $\Theta_i$ is the preferred orientation of neuron $i$. Feedforward inputs correspond to two time-varying inputs, $\phi_1(t) = C(t)\cos(\theta(t)), \phi_2 = C(t)\sin(\theta(t))$, where $C(t)$ is the stimulus magnitude and $\theta(t)$ is the stimulus orientation at time $t$ $(-\frac{\pi}{2} < \theta < \frac{\pi}{2})$. The orientation estimate , $\hat{\theta}(t)$, is decoded from the population as

$$\hat{\theta}(t) = arctan\left(\frac{\hat{\phi}_2(t)}{\hat{\phi}_1(t)}\right) \tag{21}$$

The spiking threshold includes an additional term, $\eta g_i$, that ensures that neurons with opposing preferences will not be activated to spike so easily by the excitation from opposing neurons.

Tuning curves in *Figures 6* and *7* were generated by presenting the network with a full range of stimulus orientations Each orientation was presented for 250 ms after reinitializing the network. Neuron responses were centered on their preferred orientation and the mean was taken for each subpopulation. Tuning curves after adaptation were made by lining up neuron responses to an adapting

stimulus that corresponded with its preferred orientation. Adapting stimuli were presented for 1.5 s. Standard deviations were computed on these centered data. The random gain network was identical to the above with the exception that the feedforward weight gains, $\gamma$, were randomly selected from a uniform distribution with values ranging from 3 to 9.

The tilt illusion was generated by presenting the network with an adaptor orientation (duration of 2 s) and a subsequent test orientation (250 ms). The perceived angle was decoded from the mean network output over the 250 ms presentation of the test stimulus. The adaptor had a stimulus magnitude of 25 while the test stimulus had a magnitude of 5.

# Additional information

## Funding

| Funder | Grant reference number | Author |
| --- | --- | --- |
| James S. McDonnell Foundation | | Sophie Denève |
| European Research Council | Predispike | Sophie Denève |
| Agence Nationale de la Recherche | ANR-10-LABX-0087 | Sophie Denève |

The funders had no role in study design, data collection and interpretation, or the decision to submit the work for publication.

## Author contributions

Gabrielle J Gutierrez, Conceptualization, Formal analysis, Validation, Investigation, Visualization, Writing—original draft, Writing—review and editing; Sophie Denève, Conceptualization, Resources, Supervision, Funding acquisition, Validation, Methodology, Writing—original draft, Writing—review and editing

## Author ORCIDs

Gabrielle J Gutierrez (iD) https://orcid.org/0000-0002-2350-1559

## Decision letter and Author response

Decision letter https://doi.org/10.7554/eLife.46926.013
Author response https://doi.org/10.7554/eLife.46926.014

# Additional files

## Supplementary files

• Transparent reporting form
DOI: https://doi.org/10.7554/eLife.46926.011

## Data availability

Source code for model frameworks and all data generated therefrom have been provided for all figures in our GitHub repository: https://github.com/gabrielle9/BalancedNetworkAdaptation (copy archived at https://github.com/elifesciences-publications/BalancedNetworkAdaptation).

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
