## [Decision Letter]

Thank you for submitting your article "Population adaptation in efficient balanced networks" for consideration by *eLife*. Your article has been reviewed by two peer reviewers, and the evaluation has been overseen by a Reviewing Editor and Timothy Behrens as the Senior Editor. The reviewers have opted to remain anonymous.

The reviewers have discussed the reviews with one another and the Reviewing Editor has drafted this decision to help you prepare a revised submission.

Summary:

This is a little jewel of a paper. It shows how adaptation in neurons and excitation-inhibition balance in the network can be viewed as arising from an optimisation principle. The simple cost function to be minimised is the sum of a term representing the error in signal transmission and one representing the metabolic cost of neuronal firing. The authors describe simple examples, working up to a model for orientation tuning, where they show how it naturally explains the main features of some classic observed perceptual adaptation effects. They show how understanding the phenomenon requires considering the whole local network, not just single neurons. Clearly and elegantly written, the paper was a pleasure to read, and it lays a good foundation for both further theoretical elaboration and experimental investigations.

Note that some of these are suggestions, designed to improve the paper; the authors should use their best judgment as to whether or not to include them.

1) After Equation 2, δ_j_ should be δ_ij_. (It doesn't make sense to have δ_j_ depend on i.)

2) We would suggest taking some of the inline equations and making them displayed. In particular, the equations for ŝ, r_i_, f_i_, g_i_ and Ω_ij_. As a reader, we find it much easier to find relevant variables if we don't have to crawl through lines of text.

3) The same applies to the inline equations in Materials and methods (e.g., o_i_).

4) I would also suggest putting just a little more detail of the derivation in the main text. I know these equations have been derived in a number of places, but for those who haven't memorized the derivations, even the Materials and methods will be tough to follow (especially since you switch to the multi-stimulus case). Why not just say, in the main text,

Neuron i spikes when

(s(t) – ŝ(t) – w_i)^2^ + mu sum_j_ (f_j_ + δ_ij_)^2^ <

(s(t) – ŝ(t))^2 + mu sum_j_ f_j_^2^.

Define

V_i_ = g_i_ (sum_j_ w_j_ (s_j_ – ŝ_j_) – mu f_i_),

with a spike emitted when V_i_ > 1/2, at which point it is reset to -1/2. That, and a small amount of algebra, gives you Equation 2.

5) Along the same lines, we would strongly suggest that the authors expand the description in the network model section a bit. I think many readers would find the path from Equation 3 to Equation 5 a bit magical. I know the details are there in the 2013 Boerlin et al. paper, but, because of the importance of the present paper, I think filling them in here would complete the story presented here and make its message more accessible.

6) Materials and methods has a fair number of typos (we think): it's a mix of the one stimulus case (as in the main text) and the multi-stimulus case. Please check very carefully. The things we noticed:

- above Equation 3: W_ij_ should be w_ij_.

- Equation 3: r_i_ should be f_i_.

- one of the w's should be a transpose.

7) It would help to use Greek letters to label stimulus.

Reviewer #1:

While neuronal adaptation is useful for a number of reasons, it would seem to make it hard for downstream neurons to decode responses, since they would have to know the state of adaptation. The authors provide an elegant solution: they write down a cost function that explicitly takes metabolic cost into account, then apply a technique that Sophie Deneve pioneered about a decade ago to derive the optimal network. Through magic that to this day I don't fully understand, everything works, and downstream neurons don't have to know anything about the state of adaptation to decode near optimally. On top of that, an explanation of the famous tilt illusion falls naturally out of their formalism -- something that is hard to explain by Bayesian methods.

Reviewer #2:

This is a little jewel of a paper. It shows how adaptation in neurons and excitation-inhibition balance in the network can be viewed as arising from an optimisation principle. The simple cost function to be minimised is the sum of a term representing the error in signal transmission and one representing the metabolic cost of neuronal firing. The authors describe simple examples, working up to a model for orientation tuning, where they show how it naturally explains the main features of some classic observed perceptual adaptation effects. They show how understanding the phenomenon requires considering the whole local network, not just single neurons. Clearly and elegantly written, the paper was a pleasure to read, and it lays a good foundation for both further theoretical elaboration and experimental investigations.

I would only ask the authors to expand the description in the network model section a bit. I think many readers would find the path from Equation 3 to Equation 5 a bit magical. I know the details are there in the 2013 Boerlin et al. paper, but, because of the importance of the present paper, I think filling them in here would complete the story presented here and make its message more accessible.

---

## [Author Response]

Summary:This is a little jewel of a paper. It shows how adaptation in neurons and excitation-inhibition balance in the network can be viewed as arising from an optimisation principle. The simple cost function to be minimised is the sum of a term representing the error in signal transmission and one representing the metabolic cost of neuronal firing. The authors describe simple examples, working up to a model for orientation tuning, where they show how it naturally explains the main features of some classic observed perceptual adaptation effects. They show how understanding the phenomenon requires considering the whole local network, not just single neurons. Clearly and elegantly written, the paper was a pleasure to read, and it lays a good foundation for both further theoretical elaboration and experimental investigations.Note that some of these are suggestions, designed to improve the paper; the authors should use their best judgment as to whether or not to include them.

*1) After Equation 2,* δ*_j_ should be* δ*_ij_. (It doesn't make sense to have* δ*_j_ depend on i.)*

We have made that change to agree with the convention.

*2) We would suggest taking some of the inline equations and making them displayed. In particular, the equations for* ŝ*, r_i_, f_i_, g_i_ and* Ω*_ij_. As a reader, we find it much easier to find relevant variables if we don't have to crawl through lines of text.*

These equations are now displayed on their own line.

3) The same applies to the inline equations in Materials and methods (e.g., o_i_).

Important equations in Materials and methods are now displayed.

4) I would also suggest putting just a little more detail of the derivation in the main text. I know these equations have been derived in a number of places, but for those who haven't memorized the derivations, even the Materials and methods will be tough to follow (especially since you switch to the multi-stimulus case). Why not just say, in the main text,Neuron i spikes when

*(s(t) –* ŝ *(t) – w_i)^2^ + mu sum_j_ (f_j_ +* δ*_ij_)^2^ <*

*(s(t) –* ŝ *(t))^2 + mu sum_j_ f_j_^2^.*

DefineV_i_ = g_i_ (sum_j_ w_j_ (s_j_ – ŝ_j_) – mu f_i_),with a spike emitted when V_i_ > 1/2, at which point it is reset to -1/2. That, and a small amount of algebra, gives you Equation 2.

We have included more detail in the Results section and have elaborated on the derivation in the Materials and methods.

5) Along the same lines, we would strongly suggest that the authors expand the description in the network model section a bit. I think many readers would find the path from Equation 3 to Equation 5 a bit magical. I know the details are there in the 2013 Boerlin et al. paper, but, because of the importance of the present paper, I think filling them in here would complete the story presented here and make its message more accessible.

The derivation for the network model has been expanded, particularly in the Materials and methods section.

6) Materials and methods has a fair number of typos (we think): it's a mix of the one stimulus case (as in the main text) and the multi-stimulus case. Please check very carefully.

Materials and methods section (and remainder of manuscript) was checked for typos and other errors.

The things we noticed:- above Equation 3: W_ij_ should be w_ij_.

Corrected.

- Equation 3: r_i_ should be f_i_.

Corrected.

- one of the w's should be a transpose.

We used the summation notation instead of the linear algebra expression here.

7) It would help to use Greek letters to label stimulus.

The stimulus is now labeled with the Greek letter phi.